# Perceived exertion can be lower when exercising in field versus indoors

**Karin Sofia Elisabeth Olsson[1], Ruggero Ceci[1,2], Lina Wahlgren[1], Hans Rosdahl[3], Peter Schantz[1,4]** *

1 The Research Unit for Movement, Health and Environment, Department of Physical Activity and Health, The Swedish School of Sport and Health Sciences, GIH, Stockholm, Sweden, 2 The Unit for Road Safety, Planning Department, Swedish Transport Administration, Solna, Sweden, 3 The Research Unit for Movement, Health and Environment, Department of Physiology, Nutrition and Biomechanics, The Swedish School of Sport and Health Sciences, GIH, Stockholm, Sweden, 4 Department of Public Health and Clinical Medicine, Section of Sustainable Health, Umeå University, Umeå, Sweden

* peter.schantz@gih.se

**Data Availability Statement:** The underlying research materials related to this paper are not freely and directly available because the original approval by the ethics board (The Ethics Committee North of the Karolinska Institute at the

## Abstract

### Purpose

Studies indicate that the rated perceived exertion (RPE) during physical exercise can be lower in field environments than indoors. The environmental conditions of those studies are explored. Furthermore, we study if the same phenomenon is valid when cycling indoors versus in cycle commuting environments with high levels of stimuli from both traffic and suburban-urban elements.

### Methods

Twenty commuter cyclists underwent measurements of heart rate (HR) and oxygen uptake ($\dot{V}O_2$) and RPE assessments for breathing and legs, respectively, while cycling in both laboratory and field conditions. A validated mobile metabolic system was used in the field to measure $\dot{V}O_2$. Three submaximal cycle ergometer workloads in the laboratory were used to establish linear regression equations between RPE and % of HR reserve (%HRR) and %$\dot{V}O_2$max, separately. Based on these equations, RPE from the laboratory was predicted and compared with RPE levels at the participants' individual cycle commutes at equal intensities. The same approach was used to predict field intensities and for comparisons with corresponding measured intensities at equal RPE levels.

### Results

The predicted RPE levels based on the laboratory cycling were significantly higher than the RPE levels in cycle commuting at equal intensities (67% of HRR; 65% of $\dot{V}O_2$max). For breathing, the mean RPE levels were; 14.0–14.2 in the laboratory and 12.6 in the field. The corresponding levels for legs were; 14.0–14.2 and 11.5. The range of predicted field intensities in terms of %HRR and %$\dot{V}O_2$max was 46–56%, which corresponded to median differences of 19–30% compared to the measured intensities in field at equal RPE.

Karolinska Hospital, Stockholm, Sweden (Dnr 03-637)) and the informed consent from the participants do not include such direct free access. The data will be available to all interested researchers upon request. To gain access to the data, please contact the Registrator at The Swedish School of Sport and Health Sciences, GIH, Box 5626, SE-114 86 Stockholm, Sweden, tel: +46 (0)812053700, email: registrator@gih.se.

**Funding:** This study was funded by the Public Health Funds of the Stockholm County Council (www.regionstockholm.se)(LS0401-0158)(PS) and the Research Funds of the Swedish Transport Administration (www.trafikverket.se)(TRV 2017/63917-6522 and TRV 2020/119325)(PS). The funders had no role in study design, data collection and analysis, decision to publish, or preparation of the manuscript.

**Competing interests:** The authors have declared that no competing interests exist.

## Conclusion

The cycle commuters perceived a lower exertion during their cycle commutes compared to ergometer cycling in a laboratory at equal exercise intensities. This may be due to a higher degree of external stimuli in field, although influences from other possible causes cannot be ruled out.

## Introduction

Assessment of perceived exertion during physical activities was introduced in the research already in the early 1960s by Gunnar Borg, the creator of the scale for rating the perceived exertion (RPE) [1]. That scale is constructed to have a linear relationship with heart rate (HR) at group level in the intensity range between 60–200 beats $\cdot$ min$^{-1}$ (corresponding to 6–20 in the scale) in aerobic activities involving large muscle groups such as cycling and running. Over the years, the scale has been developed with new verbal expressions and anchors, but the relations to both absolute and relative HR levels remain [2].

In the early work of Gunnar Borg, the concept of an "exertion gestalt" was formulated. It relates to how a multitude of sensations underlying the perception of exertion, e.g. muscle work, breathing, chemical substances in the blood etc., can be integrated to a perceived whole or gestalt [3]. Reviews have later addressed the origin of physiological exertion to explain the sensory cues underlying it, and its role in regulating exercise performance [e.g. 4, 5]. Both reviews argue that it is unclear how any specific sensory cue or single physiological variable can explain the perception of effort or how exertion is rated according to the RPE scale. However, in the literature underlying this assumption it is concluded that; "effort perception involves the integration of multiple afferent signals from a variety of perceptual cues" [4] from the body. On the other hand, there are indications that at least afferent feedback from the working muscles to the brain may not play a role in generating the perception of effort [6]. In line with this, a contrasting theory to the peripheral origin of effort has been developed; it suggests that the perception of exertion might originate from a copy of the central motor command, a so-called "corollary discharge" [cf. 6].

Adding to this complexity, Pennebaker and Lightner [7] suggested that the internal body signals, such as sensations of fatigue and discomfort, is in constant competition with the perception of sensory input from the environment. In their study, participants were asked to produce "comfortable pace" while running an equal distance in two different environments: a cross-country path in a wooded area, and a monotonous lap course on a flat field. The cross-country jogging, which required more focus on the external environment, elicited higher speeds (on average 10%) at comparable levels of fatigue symptoms.

To further these matters, the former RPE research group at Stockholm University, investigated a group of middle-aged men who were instructed to, in a randomized order run, (i) on a treadmill facing a wall without windows in a laboratory, and (ii) on a broad and curved outdoor recreation trail which was predominantly surrounded by greenery, had a plain surface, and was located along a lake. The field setting did not involve any ordinary traffic environment (for illustrations, see S1 Appendix) [8]. In both test conditions, participants produced three running bouts at a self-selected pace corresponding to 11, 13, and 15 on the RPE scale [1]. At the same levels of RPE, considerably higher running speeds were noted outdoors compared to in the laboratory (mean speed: 11.7 km $\cdot$ h$^{-1}$ vs indoors 7.1 km $\cdot$ h$^{-1}$, i.e. a 66% difference), and were matched with clearly higher physiological responses (HR and blood lactate). The RPE

levels were approximately 2 units lower during running in field, when equal levels of speeds and physiological measures were evaluated. Furthermore, a high reliability regarding these results was obtained between two test sessions, separated by a month, during which the participants exercised on their own according to a prescribed program [8].

Later, Mieras and colleagues [9] determined psychological and physiological responses to laboratory and outdoor cycling with recreationally trained males. In the laboratory setting, an electronically braked cycle trainer ergometer was facing a wall with no windows. The outdoor cycling was "completed along a relatively flat, out, and back course on a paved recreation trail (Keystone Trail, Omaha, NE, USA)" [9]. It follows a creek, with the surroundings being a mix of green and built-up settings for mostly commercial and retail purposes. It did not involve any ordinary traffic environments (for illustrations, see S2 Appendix). Significantly higher levels of power output (about 30%), cycling speed and HR were noted in the outdoor cycling as compared to the laboratory at similar levels of RPE.

Given the three studies mentioned that examine exercise responses in different settings [7–9], it seems that an environmental effect exists in continuous running and cycling. Thus, it could be hypothesized that various forms of physical activities produce different levels of perceived exertion depending on the environment in which they are performed. Exercising in an environment rich in visual stimulation and other sensory input may be perceived as less strenuous than an environment with poor (low) stimulation such as a laboratory or running on a monotonous field. It is, on the other hand, possible that these findings are coupled to specific effects of green environments with trees, such as reduction of stress [10], or other positive psychological effects [cf. 11]. Another possibility is that there is a connection to different hedonic valuing of a preferred versus a not preferred green setting [7], or a natural versus a, normally less preferred, synthetic setting [cf. 11].

To further understand these issues, this study compares ergometer cycling in a laboratory with commuter cycling in suburban-urban environments. Both these settings are predominantly built environments, but the commuter cycling requires attention to many traffic and urban elements, which is not the case in the laboratory. Our hypothesis is that external stimuli per se are important in masking internal cues from physical exercise, and that will lead to lower RPE levels at given exercise intensities in the suburban-urban environments compared to the laboratory.

For that purpose, 20 cycle commuters performed cycling sessions with measurements of HR and oxygen uptake ($\dot{V}O_2$) as well as assessments of RPE according to Borg's 6–20 scale [1] in both laboratory and field conditions. The laboratory session included submaximal workloads and a maximal test. In the field condition, the commuters rode their own bikes along their ordinary commuting route at a self-selected intensity, corresponding to their own normal commuting. Based on the laboratory cycling, it was possible to predict RPE levels at equal exercise intensities as in the cycle commuting in field in terms of percentages of the maximal oxygen uptake (%$\dot{V}O_2$max) and heart rate reserve (%HRR). In that way, the RPE levels rated in the field could be compared with the RPE levels predicted from the laboratory cycling. Furthermore, in a similar manner it was possible to predict which intensities that the laboratory exercise corresponded to in the field. A principle scheme of these analytical approaches is shown in Fig 1.

## Methods

The present study is a part of a greater multidisciplinary research project, Physically Active Commuting in Greater Stockholm (PACS), at the Swedish School of Sport and Health Sciences, GIH, in Stockholm, Sweden. An approval to conduct this study was obtained from the

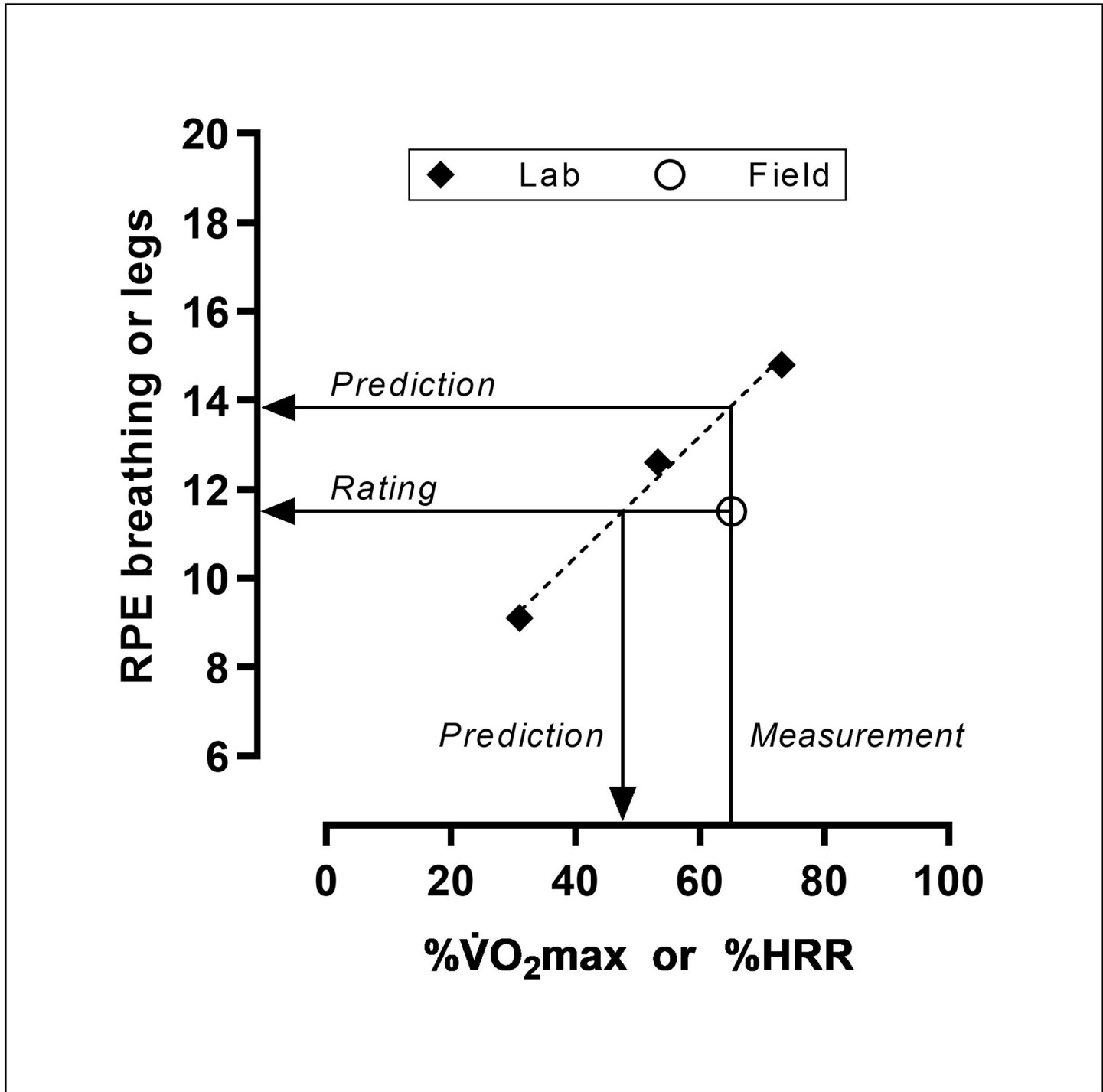

**Fig 1. A principle scheme of the analytical approach.**

Ethics Committee North of the Karolinska Institute at the Karolinska Hospital (Dnr. 03–637), Stockholm, Sweden.

## Participants

The recruitment of participants for the entire project (PACS) was initiated in 2004, and included several stages which are described in detail by Stigell and Schantz [12]. The following overall inclusion criteria were used: being at least 20 years old, living in the County of Stockholm (excluding the municipality of Norrtälje) and walking or cycling the whole way, any distance, to one's work or place of study at least once a year. A questionnaire was used to select participants, including e.g. sex, age and commuting characteristics such as modality, duration and distance. The commuting distances were measured with a criterion method based on routes drawn in maps by each respondent [13].

The present sample was selected from the single mode cyclist category, i.e. those commuters who only cycled and never walked to work. No electrically assisted bicycles were included. Further specific criteria for the present sample was that the participants had ages and commuting distances close to the overall project's median values of the male and female single mode cyclists [12]. In addition, they would also have rated their daily occupations as light or very light physically. Based on this information, ten male and ten female habitual commuter cyclists, who fulfilled the criteria, were chosen for participation. All had responded to a health declaration and certified themselves healthy for participation (individuals with high blood pressure or on medication that could affect normal HR were excluded). Prior to participation, the commuters also signed a consent of participation after receiving and reading a written information about the study procedures, and their rights as participants. This was in line with standard ethical requirements. When analysing the data, the identity of the individuals coupled to them was not disclosed to the researchers. Characteristics of the selected participants and their commuting behaviours are described in Table 1.

## Study design and standardization

The present study included two or three repeated test occasions with submaximal and maximal ergometer cycling in an exercise laboratory with a room temperature of 20˚C. Another test occasion took place in the field while each participant performed their normal daily cycle commute. Measurements of HR and $\dot{V}O_2$ as well as assessments of RPE according to Borg's 6–20 scale [1] were performed in all tests. In accordance with Ekblom and Goldbarg [14], participants were instructed to distinguish between their central perceived exertion for breathing (named breathing) and their local perceived exertion in the leg muscles (named legs). The reason for this division is that RPE for a given oxygen uptake is higher when it is executed by a small versus a large muscle group, whereas the central RPE, referred to as breathing, can be similar [14].

The first test of cycle ergometer exercise in the laboratory was carried out with a purpose of familiarization regarding all aspects of the testing, and the laboratory environment. Thereafter, six participants performed a second test occasion in the laboratory before their field tests, while the 14 remaining participants performed both a second and a third test occasion in the laboratory before their field tests. The reason for this divergence was technical problems with

**Table 1. Characteristics of the participants and their commuting behaviour (mean ± standard deviation (SD)).**

| | Age years | Height m | Weight kg | BMI kg m$^{-2}$ | HR rest beats min$^{-1}$ | Commuting frequency trips year$^{-1}$ | Commuting distance km year$^{-1}$ |
|---|---|---|---|---|---|---|---|
| **Males** (n = 10) | 43.6 ± 4.1 | 1.85 ± 0.07 | 85.0 ± 12.7 | 24.7 ± 3.0 | 58.3 ± 9.2 | 366 ± 146 | 3532 ± 1742 |
| **Females** (n = 10) | 44.0 ± 2.6 | 1.70 ± 0.05 | 65.8 ± 7.7 | 22.6 ± 2.5 | 59.6 ± 5.6 | 370 ± 117 | 2320 ± 756 |

the mobile metabolic system used, which had to be solved and the equipment re-evaluated. This delayed 14 participants' field tests by 9 to 12 months, and therefore these participants performed an extra third test in the laboratory. Due to the addition of an extra laboratory test in 14 cases, it became possible to control the stability of the RPE assessments over time by comparing laboratory tests 2 and 3. The values collected from all participants' last test occasion in the laboratory have been used as references and compared against their field tests. The mean time between the reference laboratory tests and the field tests was 13 ± 9 days (mean ± SD). A schematic illustration of the study procedure is shown in Fig 2.

Prior to all test occasions, the participants were instructed to follow the same standard guidelines. These were: 1) not to engage in any vigorous exercise for 24 hours beforehand, 2) not to cycle to the laboratory, 3) to refrain from eating, drinking, smoking and taking snuff for at least one hour before the test, 4) to not eat a large meal within three hours before the test, 5) to avoid stress, and 6) to cancel the test if they had fever, an infection or a cold.

## Equipment

**Metabolic systems.** During the laboratory tests, a stationary metabolic system (SMS), Oxycon Pro® (Carefusion GmbH, Hoechberg, Germany) was used in the mixing chamber mode for all metabolic measurements. The software used was JLAB 4.53. In the field environments, a mobile metabolic system (MMS), Oxycon Mobile, version 5.10, (CareFusion GmbH, Hoechberg, Germany) was used while measuring gas exchange variables and ventilation breath by breath. Both metabolic systems were switched on at a minimum of 30 minutes before data collection and calibrated immediately before and after each test using the automated procedures in accordance with the manufacturer's recommendations. A high precision gas of 15.00% $O_2$ and 6.00% $CO_2$ (accuracy: $O_2$ ± 0.04% and $CO_2$ ± 0.1%; Air Liquid AB, Kungsängen, Sweden) was used for calibrating the gas analyzers in both the SMS and MMS. The calibrations were always performed in the same environmental settings as the two different measurement conditions.

A facemask with non-rebreathing air inlet valves (Combitox, Dräger Safety, Lübeck, Germany) was used in both the laboratory and field conditions. All measured metabolic variables

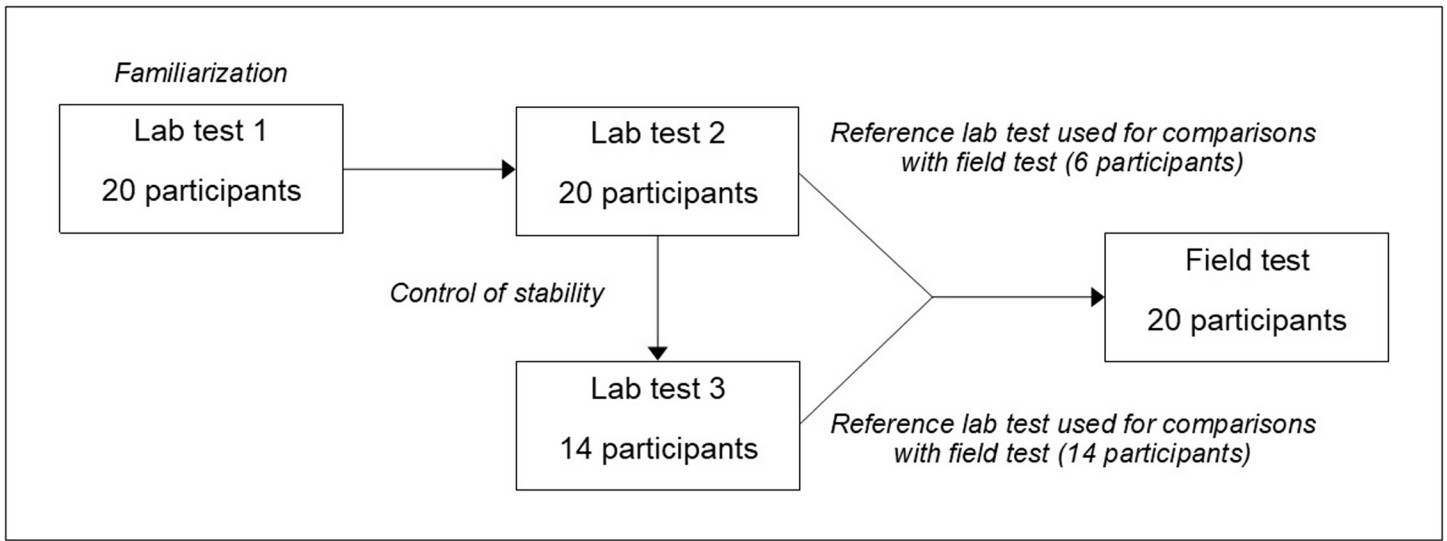

**Fig 2. Flow chart of the study procedure.**

were saved in averages of 15 seconds. Both the SMS and the MMS were carefully controlled as well as validated prior to and during the present study. This included checks with a metabolic calibrator to ensure that the two systems were interchangeable. For all methodological studies, as well as more detailed descriptions of these systems, see Rosdahl and colleagues [15], Salier Eriksson and colleagues [16], and Schantz and colleagues [17].

**Heart rate monitor.** In both the laboratory and field conditions, HR was recorded using a Polar Wearlink 31 transmitter (Polar Electro Oy, Kempele, Finland). The HR values were displayed and saved in averages of 15 seconds and stored in the SMS or the MMS. For safety reasons, HR values were also stored in averages of 15 seconds in the Polar Electro S610i HR monitor watch (Polar Electro Oy, Kempele, Finland) during the field tests. However, the HR values were always used from the MMS, except for three periods (1–6.25 minutes), one in each of three participants, when values were missing due to technical problems. In these cases, the HR values from the watch substituted the missing values. To confirm that this was appropriate, all individuals' watch HR values were compared with the corresponding HR values from the MMS. No significant differences were detected in this comparison.

**Cycle ergometer.** A mechanically braked pendulum cycle ergometer (828E Monark Exercise AB, Vansbro, Sweden) was used for performing the cycle exercise in the laboratory. Before each test, the scale was zeroed while each participant sat on the saddle with his or her feet resting on the frame between the pedals. The saddle height was adjusted so that one knee of the participant was slightly bent when the foot was on the pedal in its lowest position. A digital metronome (DM70 Seiko S-Yard Co. Ltd, Tokyo, Japan) was used to instruct the participants to maintain correct cycling cadence. The workload was controlled every minute by checking the cadence and the braking force as indicated on the pendulum position.

## Procedure

**Laboratory tests.** All the repeated tests in the laboratory kept the same standard procedures of measurements. Firstly, a few measurements at rest were carried out before the cycle ergometer exercise. This included measurements of body weight and height as well as resting HR. For determining the resting HR, the participants rested quietly in supine position on a treatment table for ten minutes. The values from the last five minutes were averaged and determined as the resting HR.

The submaximal cycle ergometer exercise consisted of three different workloads; 50, 100 and 150 watt (W) for the females, and 100, 150 and 200 W for the males. The cycling cadence was kept to 50 revolutions per minute (rpm) in accordance with Åstrand [18, p. 19]. For each workload, the participants cycled until steady state HR was attained for two consecutive minutes (total work time approximately six minutes), after which the resistance was increased. Steady state HR was defined as the average HR value of one minute, based on the 30 seconds before and the 30 seconds after each full minute, were 2–3 beats · min$^{-1}$ at two consecutive minutes. The third workload was increased to only 125 W or 175 W for the females and males, respectively, if, after the second workload, the participant's HR was higher than 150 beats · min$^{-1}$ and their rated RPE exceeded 15 [1].

Between the second and third workloads, the participants sustained pedalling for one minute at a self-chosen low cadence, and with a low resistance of 5 Newton (N). They were then instructed to resume the cadence of 50 rpm while the investigator gradually increased the resistance until, after one minute, the third workload was achieved (resistance was increased to 50 W during the first 15 seconds, to 100 W the next 15 seconds and successively to the final workload during the last 30 seconds). During all submaximal workloads, the participants sat in an upright position with their hands laying on the handlebars. The RPE assessments for breathing

and legs, respectively, took place during the latter part of the final minute of each workload. After this phase of submaximal exercise, and before the maximal exercise, a period of two minutes followed by continued cycling with a self-chosen low cadence at a resistance of 5 N.

The maximal exercise was carried out with a cycling cadence of 80 rpm, as this cadence has been shown to be associated with the longest time to exhaustion at maximal efforts [19]. During the first three minutes, the workloads were set to 60, 100, and 120 or 140 W for each minute. The latter alternatives depended on which third workload the participants had performed during the submaximal exercise. Thus, 120 W was selected if the third level had been 125 W or 175 W for the females and males, respectively, whereas 140 W was used if it had been 150 W or 200 W. After these initial three minutes, the resistance increased by 20 W every minute until voluntary exhaustion occurred and ended the test. The RPE assessments for breathing and legs, respectively, were made immediately after this completion. To ensure that the participants achieved their $\dot{V}O_2$max, at least two of the following criteria were met by each participant: 1) a plateau in $\dot{V}O_2$ despite increasing exercise intensity (defined as a $\dot{V}O_2$ increment of $< 150$ mL $\cdot$ min$^{-1}$), 2) a respiratory exchange ratio of $\geq 1.1$, and 3) a RPE of $\geq 17$ [1, 20, 21]. During all cycle ergometer exercise, the participants looked into a shield with a blue-grey colour without any pattern. Thus, the visual stimulation was at a low level.

**Field tests.** The participants pedaled their own bicycles either to or from their workplace choosing themselves which direction and time suited them best. They were instructed to cycle at their ordinary commuting intensities for the entire route. Characteristics of the commuters' bicycles, such as weight and number of gears, can be found in Schantz and colleagues [22]. The tests were carried out between June and November. 18 of the tests were performed during the morning rush hours, while the remaining two were conducted during the evening rush hours. The cycle commutes took place in the inner urban and suburban areas of Greater Stockholm, Sweden. This mixture of areas meant that the commuters experienced a variety of route environments, in terms of e.g. buildings, greenery, topography and traffic. The study setting for the field tests, in overall terms, is illuminated in S3 Appendix. For a more detailed description of these areas, see Wahlgren and Schantz [23].

The participants were met at their designated start address by one of the investigators who transported the measurement equipment. Prior to the commutes, the MMS was placed in a custom-made backpack on the participants. A GPS was also placed in the backpack to track the road. The starting time of the cycle commutes was synchronized with a second investigator waiting at the final address. When the commuters arrived at their destinations, the total trip time was noted (means were; 29 minutes for men and 23 for women). The participants were then immediately asked to rate RPE for breathing and legs, respectively, of the overall commute. They also stated the amount of stops they made at traffic lights as well as other stops [22]. Levels of ambient conditions in terms of temperature, relative humidity and wind speed for each individual cycle commute were obtained from the website of the Stockholm-Uppsala Air Quality Management Association [24], see Table 2.

**Assessments of the cycle route environments.** The Active Commuting Route Environment Scale (ACRES) was used for assessments of the participants' perceptions and appraisals of their route environments. For most of the participants, this was done a year before the field

**Table 2. Ambient conditions during the cycle commuting in field (mean ± SD).**

| | Temperature (˚C) | Relative humidity (%) | Wind speed (m $\cdot$ s$^{-1}$) |
|---|---|---|---|
| **Males** (n = 10) | 10 ± 4 | 77 ± 12 | 4.0 ± 2.0 |
| **Females** (n = 10) | 12 ± 4 | 71 ± 22 | 4.4 ± 1.9 |

tests. ACRES has been characterized by considerable criterion-related validity and reasonable test-retest reproducibility [23, 25]. In this study, two items from ACRES were used for overall assessments of the routes cycled to work. The first was whether the route environments inhibit or stimulate the cycling. This item was formulated as follows: "Do you think that, on the whole, the environment you cycle in stimulates/inhibits your commuting?". The second item was whether the route environment was experienced as unsafe or safe for reasons of traffic. This was formulated as follows: "How unsafe/safe do you feel in traffic as a cyclist along your route?". The items' response scales ranged between 1 and 15, with adjectival opposites labelled as "inhibits a lot" and "stimulates a lot", respectively, "very unsafe" and "very safe". Along the scale, the position number 8 represented a neutral position labelled as neither nor e.g. inhibiting or stimulating. The questionnaire instructions included a drawn map that separated the areas into; inner urban and suburban–rural areas [25]. Participants were asked to differentiate their commuting experiences between these two areas, see the ratings in Table 3.

## Analytical approach

The individual levels of RPE, HR and $\dot{V}O_2$, obtained from the reference laboratory test, have been applied as a basis for the comparisons with the rated RPE values in field. Moreover, analyses of the stability of the RPE assessments over time in laboratory conditions have also been conducted by comparing individual predictions from test 2 and test 3 (based on those 14 participants who performed an extra third laboratory test) (see Fig 2).

At first, paired HR and $\dot{V}O_2$ values from the last two consecutive minutes at steady state in each of the three submaximal cycle ergometer workloads, at laboratory tests 2 and 3, were averaged for each individual and used for further analyses. The maximal values from the reference laboratory test were also used and determined by averaging the highest consecutive paired values of HR and $\dot{V}O_2$ during one minute. These maximal values, as well as the resting HR values from the reference laboratory test, were used in all calculations of individual exercise intensities in terms of %HRmax, %HRR and %$\dot{V}O_2$max. These relative intensities were used to describe the three submaximal workloads at the reference laboratory test as well as the mean levels of exercise intensities during the cycle commutes in field.

For the comparisons of RPE predictions between test 2 and test 3, the absolute HR and $\dot{V}O_2$ values from the three submaximal cycle ergometer workloads in these tests were calculated into exercise intensities in terms of %HRR and %$\dot{V}O_2$max. These relative intensities were used given possible interindividual differences in training status and maximal HR levels [cf. 2]. Linear regression equations were calculated based on these three intensity levels, and their corresponding RPE levels for breathing and legs, respectively. Thus, this led to a total of eight regression equations for each participant: %HRR-RPE breathing, %HRR-RPE legs, %$\dot{V}O_2$max-RPE breathing, and %$\dot{V}O_2$max-RPE legs, for test 2 and test 3, respectively. Thereafter,

**Table 3. Route environment ratings in the inner urban and suburban–rural areas of Greater Stockholm, Sweden (mean ± SD).**

| | Inner urban area (n = 7 males, 9 = females) | | Suburban–rural area (n = 9 males, 7 = females) | |
|---|---|---|---|---|
| | Inhibits -stimulates cycling | Unsafety -safety traffic | Inhibits -stimulates cycling | Unsafety -safety traffic |
| **Males** | 9.1 ± 4.4 | 5.6 ± 3.0 | 11.6 ± 1.7 | 10.8 ± 3.4 |
| **Females** | 9.2 ± 4.0 | 7.2 ± 3.6 | 10.9 ± 3.0 | 10.9 ± 2.4 |

Note: 1 = "inhibits cycling a lot" or "very unsafe for reasons of traffic"; 15 = "stimulates cycling a lot" or "very safe for reasons of traffic". 8 = a neutral position labelled as neither nor e.g. inhibiting or stimulating.

the three individual levels of exercise intensities for %HRR and %$\dot{V}O_2$max, respectively, from the reference laboratory test were used in their corresponding regression equation to predict RPE values for each test occasion. Thus, the same levels of intensities were used in the regression equations for both test 2 and test 3. The individual levels of absolute RPE differences between test 2 and test 3 were calculated for all comparisons.

Regarding the laboratory and field comparisons, the RPE levels and exercise intensities for %HRR and %$\dot{V}O_2$max, respectively, from the three submaximal cycle ergometer workloads at the reference laboratory test were used to establish four linear regression equations for each individual: %HRR-RPE breathing, %HRR-RPE legs, %$\dot{V}O_2$max-RPE breathing, and %$\dot{V}O_2$max-RPE legs. The individual mean exercise intensities of the cycle commutes were then used in the corresponding regression equations to predict RPE values representing the overall cycle commutes in field. The absolute differences between the predicted and rated RPE were calculated for all individuals' comparisons. In the same way, the average relative commuting intensities for each individual were predicted by inserting the corresponding rated RPE values of the overall cycle commutes, into each of the four above-mentioned regression equations. The individual absolute and relative differences between the measured and predicted commuting intensities were calculated.

## Statistical analyses

The Statistical Package for the Social Sciences (SPSS, 27.0, Chicago, IL, USA) was used to perform the statistical analyses. Illustrations were created with GraphPad Prism® 8.0 software package (GraphPad Software Inc., San Diego, CA, USA). An alpha level of 0.05 was used to determine statistical significance. In cases when the same rated RPE or measured intensity levels in the field were compared twice with different predictions of RPE or intensity, a Bonferroni correction for multiple adjustments was applied to reduce the Type I error probability [26, p. 377]. To keep the chosen alpha level (0.05) consistent across all comparisons, the P-values obtained were multiplied by two instead of lowering the alpha value. Values are reported as mean ± SD, unless otherwise stated.

The normality of distribution of all absolute and relative differences were evaluated with the Shapiro-Wilk test. Because the absolute differences between the predicted and rated RPE levels in the field were not normally distributed in two of four cases, all of these differences were analysed using both the parametric one-sample T-test and the non-parametric one-sample Wilcoxon signed rank test. However, since the two different tests generated similar significance levels, only the P-values from the one-sample T-test are reported. In the cases with absolute and relative differences between measured and predicted commuting intensities, the differences were not normally distributed in five of eight cases. Therefore, the one-sample Wilcoxon signed rank test was used in all these cases. The relative intensity differences were expressed as median values, instead of means, due to the occurrence of outliers. Confidence intervals (CI) of 95% were calculated for all predicted and rated RPE levels, as well as for the measured and predicted commuting intensities. Linear regressions were used to predict RPE values.

## Results

### Exercise characteristics in the laboratory

RPE and exercise intensities, in terms of absolute and relative levels of HR and $\dot{V}O_2$, from the submaximal and maximal cycle ergometer exercise at the reference laboratory test, are given in Table 4. For the males, the three submaximal work rates induced mean levels of RPE ranging

**Table 4. RPE, HR and $\dot{V}O_2$ during submaximal and maximal cycle ergometer exercise in the laboratory (mean ± SD).**

| Work rates | | RPE | | HR | | | $\dot{V}O_2$ | |
|---|---|---|---|---|---|---|---|---|
| | W | breathing | Legs | beats · min⁻¹ | %HRmax | %HRR | L · min⁻¹ | %$\dot{V}O_2$max |
| **Males** (n = 10) | | | | | | | | |
| **Work rate 1** | 100 ± 0 | 10.3 ± 1.2 | 10.2 ± 1.0 | 98 ± 9 | 56.5 ± 3.4 | 34.5 ± 3.5 | 1.45 ± 0.15 | 36.7 ± 4.1 |
| **Work rate 2** | 150 ± 0 | 12.7 ± 1.6 | 12.2 ± 1.5 | 118 ± 10 | 67.7 ± 3.2 | 51.4 ± 4.1 | 2.09 ± 0.15 | 52.9 ± 5.6 |
| **Work rate 3** | 193 ± 12 | 14.2 ± 2.1 | 14.3 ± 1.8 | 137 ± 12 | 78.7 ± 4.0 | 68.0 ± 5.6 | 2.65 ± 0.22 | 67.0 ± 6.9 |
| **Max** | | 17.8 ± 1.4 | 18.3 ± 1.2 | 174 ± 7 | 100.0 ± 0.0 | 100.0 ± 0.0 | 3.99 ± 0.55 | 100.0 ± 0.0 |
| **Females** (n = 10) | | | | | | | | |
| **Work rate 1** | 50 ± 0 | 8.9 ± 1.9 | 9.1 ± 1.7 | 97 ± 8 | 55.0 ± 3.0 | 31.7 ± 4.1 | 0.81 ± 0.09 | 31.1 ± 3.9 |
| **Work rate 2** | 100 ± 0 | 12.6 ± 1.1 | 12.6 ± 1.5 | 121 ± 9 | 68.9 ± 4.3 | 52.9 ± 6.1 | 1.39 ± 0.12 | 53.3 ± 6.5 |
| **Work rate 3** | 140 ± 13 | 14.6 ± 0.8 | 14.8 ± 1.5 | 145 ± 9 | 82.9 ± 4.3 | 73.9 ± 6.7 | 1.91 ± 0.16 | 73.2 ± 7.3 |
| **Max** | | 18.3 ± 1.5 | 17.9 ± 1.2 | 175 ± 10 | 100.0 ± 0.0 | 100.0 ± 0.0 | 2.63 ± 0.32 | 100.0 ± 0.0 |

between 10.3–14.2 (breathing) and 10.2–14.3 (legs). The corresponding levels for the females varied between 8.9–14.6 (breathing) and 9.1–14.8 (legs) (Table 4).

## Stability controls of RPE levels in the laboratory

Stability controls of RPE over time are presented in Table 5 (n = 14). Comparisons are made between cycle ergometer test 2 and test 3 for predicted RPE levels based on the regression equations for %HRR-RPE breathing, %HRR-RPE legs, %$\dot{V}O_2$max-RPE breathing, and %$\dot{V}O_2$max-RPE leg. No significant RPE differences were found. The range of all absolute differences was -0.7 to 0.2 RPE units (Table 5).

## Exercise characteristics in the field

Descriptive characteristics of the cycle commutes, such as duration, distance, speed, and cycling environment are reported in Table 6 for males and females, respectively. The overall RPE levels as well as the average exercise intensities, in terms of both absolute and relative HR and $\dot{V}O_2$ levels, of these cycle commutes are given in Table 7. For the males, the average RPE was 12.8 for breathing and 11.5 for legs. The corresponding levels for the females were 12.4 (breathing) and 11.5 (legs) (Table 7).

## Comparisons of RPE levels between laboratory and field conditions

All participants' levels of RPE and exercise intensities of the three submaximal cycle ergometer work rates at the reference laboratory test (cf. Table 4), as well as the corresponding levels from the cycle commutes in field (cf. Table 7), are illustrated in Fig 3.

 Comparisons of the predicted RPE levels in field, based on the cycle ergometer exercise at the reference laboratory test, and the rated RPE values of the overall cycle commutes in field are presented in Table 8 for all participants. In all comparisons, the laboratory based predicted RPE levels were significantly higher than the RPE rated in field. Based on linear regression equations established between %HRR and RPE, the absolute mean differences were: 1.6 RPE units (P < 0.01) for breathing and 2.7 (P < 0.001) for legs. The corresponding RPE predictions based on %$\dot{V}O_2$max yielded mean differences of: 1.4 RPE units (P < 0.05) for breathing and 2.5 (P < 0.001) for legs (Table 8).

**Table 5. Comparisons of predicted RPE levels between test 2 and test 3 in the laboratory for males and females together (n = 14) (mean ± SD and (95% CI)).**

| Work rate intensities | RPE Breathing | | | RPE Legs | | |
|---|---|---|---|---|---|---|
| | Test 2 | Test 3 | Absolute difference P-value | Test 2 | Test 3 | Absolute difference P-value |
| *%HRR* | | | | | | |
| **Work rate 1** | 9.1 ± 2.3 | 9.8 ± 1.5 | -0.7 ± 2.0 | 9.2 ± 2.5 | 9.8 ± 1.5 | -0.6 ± 2.1 |
| 33.7 ± 4.0% | (7.7–10.4) | (9.0–10.7) | 0.181 | (7.8–10.7) | (9.0–10.7) | 0.311 |
| **Work rate 2** | 11.7 ± 2.1 | 12.2 ± 1.4 | -0.5 ± 1.5 | 12.0 ± 2.1 | 12.3 ± 1.5 | -0.3 ± 1.7 |
| 52.4 ± 4.6% | (10.5–12.9) | (11.4–13.0) | 0.256 | (10.8–13.2) | (11.5–13.2) | 0.473 |
| **Work rate 3** | 14.5 ± 1.8 | 14.6 ± 1.7 | -0.2 ± 1.4 | 14.9 ± 1.6 | 14.9 ± 1.7 | 0.0 ± 1.7 |
| 72.0 ± 7.0% | (13.4–15.5) | (13.7–15.6) | 0.694 | (14.0–15.9) | (13.9–15.9) | 0.963 |
| *%$\dot{V}O_2$max* | | | | | | |
| **Work rate 1** | 9.1 ± 2.1 | 9.8 ± 1.5 | -0.7 ± 1.7 | 9.3 ± 2.3 | 9.8 ± 1.5 | -0.5 ± 1.9 |
| 34.0 ± 4.2% | (7.9–10.3) | (8.9–10.6) | 0.176 | (8.0–10.6) | (9.0–10.7) | 0.337 |
| **Work rate 2** | 12.0 ± 2.0 | 12.3 ± 1.4 | -0.3 ± 1.1 | 12.2 ± 1.9 | 12.4 ± 1.5 | -0.2 ± 1.4 |
| 52.5 ± 4.1% | (10.8–13.2) | (11.4–13.1) | 0.360 | (11.1–13.4) | (11.6–13.3) | 0.657 |
| **Work rate 3** | 14.7 ± 2.0 | 14.6 ± 1.7 | 0.1 ± 1.3 | 15.1 ± 1.6 | 14.8 ± 1.7 | 0.2 ± 1.5 |
| 70.2 ± 8.0% | (13.5–15.8) | (13.6–15.6) | 0.841 | (14.1–16.0) | (13.9–15.8) | 0.622 |

Note: The RPE predictions are based on individual linear regression equations established between %HRR-RPE and %$\dot{V}O_2$max-RPE for breathing and legs, respectively, at test 2 and test 3.

Calculations of absolute differences = "Test 2"–"Test 3".

## Comparisons of intensity levels between laboratory and field conditions

Comparisons of the measured relative intensities in the field in terms of %$\dot{V}O_2$max and %HRR and the corresponding predicted intensities, based on the relationships between intensities and RPE levels obtained during the cycle ergometer exercise at the reference laboratory test, are presented in Table 9 for all participants. In both absolute and relative terms, all comparisons between the measured and predicted intensities, yielded significantly higher measured intensities compared to the predicted levels. For instance, the median values of all relative differences ranged between 19 and 30% (range of P-values: < 0.001–0.025).

## Discussion

To our knowledge, no other study has compared cycle exercise induced RPE levels in a laboratory setting versus a cycling commuting route environment at equal exercise intensities in terms of %HRR and %$\dot{V}O_2$max. The cycle exercise in the laboratory was performed on an ergometer cycle, whereas the cycling in field was undertaken using each participant's own bicycle. The main findings were that the predicted RPE levels from the ergometer cycling in the laboratory (means: 14.0–14.2) were significantly higher in all cases compared to the rated

**Table 6. Characteristics of the cycle commuting in field (mean ± SD).**

| | Duration min | Distance km | Speed km · h⁻¹ | Cycling environment |
|---|---|---|---|---|
| **Males** (n = 10) | 29.1 ± 7.2 | 9.61 ± 2.20 | 20.1 ± 2.7 | 1.10 ± 0.32 |
| **Females** (n = 10) | 23.2 ± 5.0 | 6.51 ± 1.51 | 16.8 ± 1.9 | 0.90 ± 0.57 |

Note: Cycling environment in Greater Stockholm, Sweden: 0 = inner urban; 1 = inner urban-suburban; 2 = suburban.

**Table 7. RPE, HR and $\dot{V}O_2$ of the overall cycle commuting in field (mean ± SD).**

| | RPE | | HR | | | $\dot{V}O_2$ | |
|---|---|---|---|---|---|---|---|
| | breathing | legs | beats · min⁻¹ | %HRmax | %HRR | L · min⁻¹ | %$\dot{V}O_2$max |
| **Males** (n = 10) | 12.8 ± 1.0 | 11.5 ± 1.1 | 136 ± 13 | 78.5 ± 8.3 | 67.6 ± 12.6 | 2.61 ± 0.57 | 65.3 ± 11.3 |
| **Females** (n = 10) | 12.4 ± 2.0 | 11.5 ± 2.5 | 137 ± 7 | 78.0 ± 3.6 | 66.7 ± 4.1 | 1.70 ± 0.21 | 65.1 ± 10.0 |

RPE levels in the field (means: 11.5–12.6) at the same measured intensities in field. These overestimates amounted to averages of 1.4–1.6 RPE units for breathing, and 2.5–2.7 RPE units for legs. When instead comparing the predicted field intensities with the corresponding measured field intensities at equal RPE levels, it was found that the measured intensities were significantly higher in all cases. This resulted in relative differences, expressed as medians, ranging between 19 and 30%.

These results are in line with three previous studies undertaken in different field conditions [7–9]. To facilitate an overview of all four studies, relevant data has been compiled in S4 Appendix. Overall, these studies indicate that exercise performed in environments with high levels of visual stimuli is perceived as less exerting compared to exercising in environments with lower levels of stimulation, such as in a laboratory or running on a monotonous field. One explanation could be that the visual stimuli act to some extent through a passive fascination. Alternatively, the stimuli from the external environment during commuter cycling could demand a focused attention to analyze the traffic conditions. Thus, although the cause for the findings cannot be determined, our hypothesis that external stimuli per se are important for masking internal cues from physical exercise is supported by the lower RPE levels when cycling in the suburban-urban environments compared to the laboratory.

The two conditions compared in the present study are, however, different in both methodological and environmental aspects. Therefore, potential moderating factors of the RPE levels need to be scrutinized in relation to the present results.

## Potential moderators of RPE

The submaximal cycle ergometer exercise in the laboratory included three constant workloads. During these, the participants were instructed to cycle with a fixed cadence of 50 rpm for approximately 6 minutes per workload. This contrasts with the cycling commuting in field, during which the participants pedalled their own bikes along their ordinary routes at a self-selected intensity and cadence, corresponding to their own normal commuting.

**Cycling cadence.** With regard to the cycling cadence, even if this variable was not monitored in the field, it can be assumed that varying cadences have been used in the different measurement conditions. Most likely, the participants preferred to use an average higher cadence in the field compared to the fixed level of 50 rpm in the laboratory [27]. Hansen and colleagues [27] studied a range of cadences around 60–90 rpm at two different cycle ergometer workloads covering our average commuting intensity (65% of $\dot{V}O_2$max; Table 7). Based on their results, the relationship between RPE and $\dot{V}O_2$ does not appear to be affected by using different cadences. This is supported by Kounalakis and colleagues [28], who found no significant differences in RPE or HR at the same absolute $\dot{V}O_2$ values when comparing cadences of 40 and 80 rpm during 90 minutes of cycle ergometer exercise. On the contrary, an early study [29] noted that a cycling cadence of 40 rpm compared to 60 and 80 rpm generated higher RPE levels despite no differences in $\dot{V}O_2$. These findings were most evident at the higher workloads (corresponding to approximately 45 and 64% of $\dot{V}O_2$max). Given this discrepancy in findings,

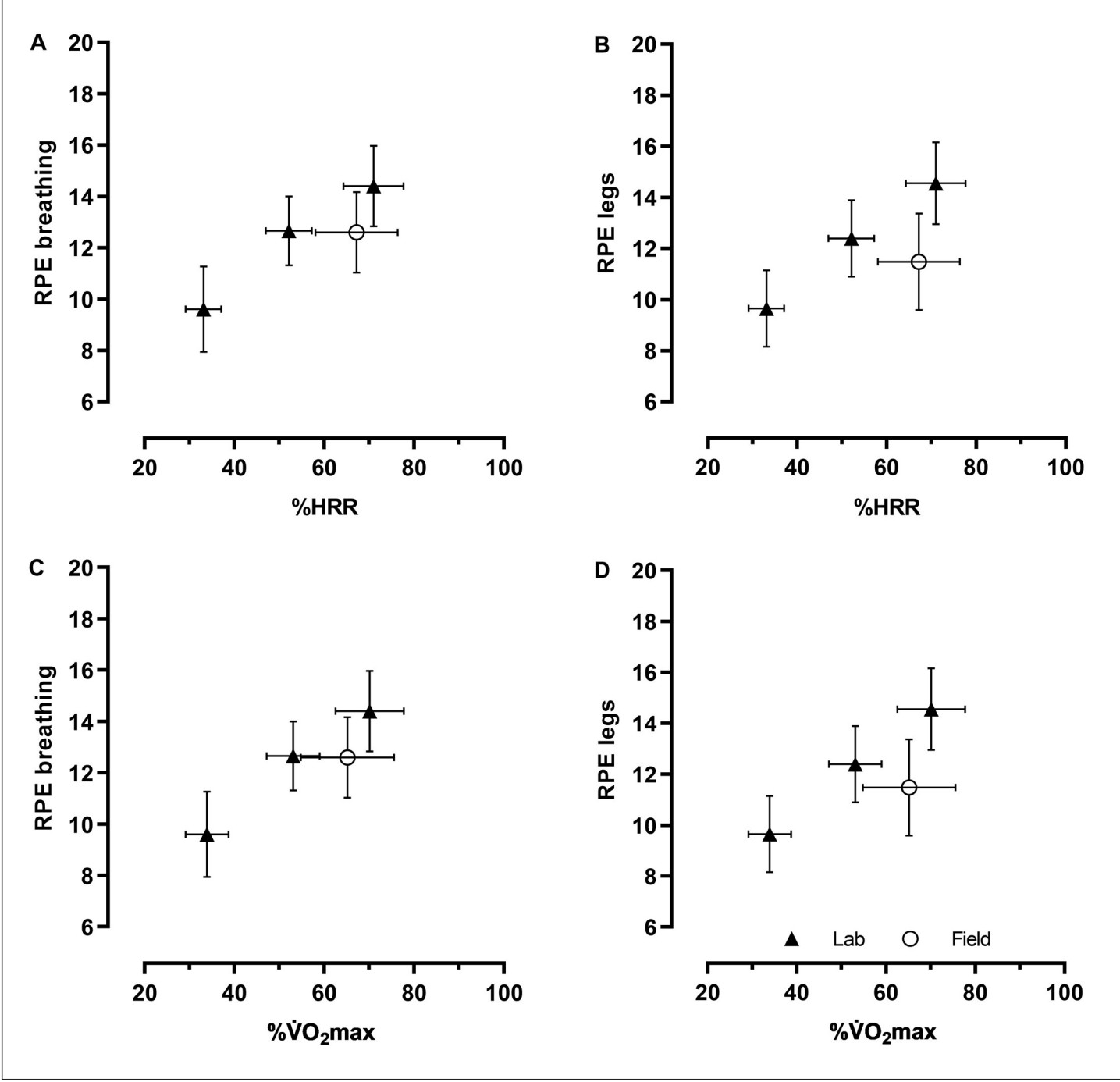

**Fig 3. Levels of RPE and exercise intensities of the three submaximal cycle ergometer work rates at the reference laboratory test as well as the corresponding levels from the cycle commutes in field.** Based on all participants' individual values (n = 20), and presented as mean ± SD. A) RPE breathing and %HRR, B) RPE legs and %HRR, C) RPE breathing and %$\dot{V}O_2$max, and D) RPE legs and %$\dot{V}O_2$max.

it cannot be ruled out that a use of varying cadences has contributed to the present deviations in RPE between the laboratory and field conditions.

**Table 8. Comparisons between laboratory based predicted and rated RPE levels for the overall cycle commuting in field for all participants (n = 20) (mean ± SD and (95% CI)).**

| Field: measured intensity | RPE Breathing | | | RPE Legs | | |
|---|---|---|---|---|---|---|
| | Laboratory: field prediction | Field: rating | Absolute difference P-value | Laboratory: field prediction | Field: rating | Difference (Lab–Field) P-value |
| **%HRR** | 14.2 ± 2.0 | 12.6 ± 1.6 | 1.6 ± 2.0 | 14.2 ± 2.0 | 11.5 ± 1.9 | 2.7 ± 2.4 |
| 67.2 ± 9.2% | (13.3–15.2) | (11.9–13.3) | 0.004 | (13.3–15.1) | (10.6–12.4) | < 0.001 |
| **%V̇O₂max** | 14.0 ± 2.0 | 12.6 ± 1.6 | 1.4 ± 2.2 | 14.0 ± 2.1 | 11.5 ± 1.9 | 2.5 ± 2.6 |
| 65.2 ± 10.4% | (13.1–15.0) | (11.9–13.3) | 0.018 | (13.1–15.0) | (10.6–12.4) | 0.001 |

Note: The RPE predictions are based on individual linear regression equations established between %HRR-RPE and %V̇O₂max-RPE for breathing and legs, respectively, at the reference laboratory test.

Calculations of absolute differences = "Laboratory: field prediction"–"Field: rating".

P-values have been adjusted for multiple comparisons according to the Bonferroni correction.

Interestingly, numerically higher RPE mean differences between the two measurement conditions were noted in the rating for legs (2.5–2.7 RPE units) compared to breathing (1.4–1.6 RPE units). Possibly, these different results may be due to that varying cadences have been used. According to the two-factor categorization of perceived exertion [14], it is likely that the participants perceived that the local muscular strain in their legs affected them more differently, than their central breathing exertion did, when using varying cadences. Nevertheless, the fact that RPE for breathing differs significantly between the two conditions indicates that there may be more moderating factors affecting the present RPE results than the cycling cadence.

**Exercise duration.** The commuting durations were about 29 and 23 minutes for males and females, respectively (Table 6). These are slightly longer time periods than the total exercise duration for the submaximal cycle ergometer exercise in the laboratory (about 18 minutes). Due to these time variations, it is possible that the present RPE differences between the laboratory and the field environment have been underestimated, as RPE has been shown to rise with increasing exercise duration of ergometer cycling and treadmill walking at work rates corresponding to 60% of V̇O₂max [30, 31].

**Table 9. Comparisons between laboratory based predicted and measured average intensity levels for cycle commuting in field for all participants (n = 20) (mean ± SD and (95% CI)).**

| Field: measured intensity | Intensities are based on RPE Breathing | | | Intensities are based on RPE Legs | | |
|---|---|---|---|---|---|---|
| | Laboratory: field prediction | Absolute difference P-value | Relative difference* P-value | Laboratory: field prediction | Absolute difference P-value | Relative difference* P-value |
| **%HRR** | 55.8 ± 17.8 | 11.4 ± 18.2 | 22.0 | 45.9 ± 17.7 | 21.3 ± 18.3 | 30.0 |
| 67.2 ± 9.2% | (47.5–64.2) | 0.004 | 0.002 | (37.6–54.2) | < 0.001 | < 0.001 |
| **%V̇O₂max** | 56.4 ± 16.1 | 8.8 ± 18.1 | 19.1 | 46.9 ± 17.6 | 18.3 ± 17.4 | 22.2 |
| 65.2 ± 10.4% | (48.9–63.9) | 0.041 | 0.025 | (38.6–55.1) | 0.001 | < 0.001 |

Note: The intensity predictions are based on individual linear regression equations established between %HRR-RPE and %V̇O₂max-RPE for breathing and legs, respectively, at the reference laboratory test. Calculations of absolute differences = "Field: measured intensity"–"Laboratory: field prediction".

Calculations of relative differences = ("Absolute difference" · "Laboratory: field prediction"$^{-1}$) · 100.

*The relative differences are expressed as median values.

P-values have been adjusted for multiple comparisons according to the Bonferroni correction.

This presumed underestimation of RPE difference between the two conditions, is further supported by data from 1661 commuter cyclists of both sexes, and with varying ages. Multiple regression analyses showed that Borg's 6–20 RPE scale increased with 0.58 RPE units per 10 minutes at a given cycling speed when taking speed, duration, age and sex into account (Peter Schantz, personal communication 2022-03-22). Furthermore, given that longer exercise durations are related to higher cycling speeds in the above mentioned large sample of cyclists [32], it can be anticipated that they who cycle longer durations have higher $\dot{V}O_2$max. Thereby, they can sustain a given cycling speed at a lower % of $\dot{V}O_2$max, and at a lower RPE as well. Thus, it is possible that the value from the multiple regression analyses stated above represents an additional form of underestimation of the duration effect on the present RPE results.

**Constant versus variable exercise.** Given the naturalistic setting of the field tests in sub-urban-urban commuting environments, including varying topography and different traffic situations such as queuing, turning and red lights, etc., it is clear that the field cycling involved distinctly more variable and intermittent exercise compared to the constant submaximal cycle ergometer workloads in the laboratory. The question is whether this affects the relation between RPE and the overall workload responses in terms of %HRR and %$\dot{V}O_2$max. Based on previous studies, it appears that the overall workload during a given exercise duration, regardless of whether the intensity levels are constant or varying intermittently, is the important determinant of RPE in this respect [33, 34]. In such a case, the RPE responses in the laboratory and field conditions are comparable at equal mean exercise intensities. However, more studies of these issues are warranted.

**Instructions and protocols.** The fact that the participants self-regulated their commuting cycling in field (regarding e.g. levels of cycling gears, cadence, and exercise intensity) versus that they followed a protocol in the laboratory, is also a methodological difference possibly affecting RPE. This is because a high degree of autonomy during high-intensity interval training as well as during submaximal exercise sessions has been associated with significantly lower RPE levels compared to conventional and prescribed exercise protocols [35, 36]. Thus, it is possible that also the present results can be due to differences in exercise autonomy.

Moreover, the participants rated RPE during the last minute of each of the submaximal workloads in the laboratory, whereas in the field setting, they made their RPE assessments for the total commute immediately thereafter. Since RPE assessments for longer exercise durations have been shown to reflect the exertion rate close to the end of the exercise rather than the total exercise period [37], the present relations between RPE and exercise intensities in the field may be questioned. Given this, we have compared mean values of %HRR and %$\dot{V}O_2$max, respectively, between the entire cycle commutes and the last 5-min periods for all participants in this study (see S5 Appendix). These analyses were based on a data set where the transition periods at both the start and the end of the cycle commutes have been excluded, i.e. the transitions from resting level to exercise intensity as well as from exercise intensity back to resting level. The comparisons showed that there were no differences in terms of % of $\dot{V}O_2$max between the entire cycle commutes and the last 5-min periods. On the other hand, % of HRR was slightly lower for the entire commutes compared to the last 5-min periods (averages: 68.9% versus 71.5%; P-value = 0.003; S5 Appendix). Thus, in line with Kilpatrick and colleagues [37], the present RPE ratings in the field may correspond to a somewhat higher mean level of %HRR than the one stated in this study. However, this would have led to an increased difference between the laboratory based predicted RPE and the rated RPE in the field, and therefore, it cannot explain the present results.

**Environmental aspects.** The individual cycle commuting routes were in the inner urban and/or suburban areas of Greater Stockholm. Thereby, the participants experienced a variety of

visual and auditory stimuli from the environment in terms of e.g. buildings, greenery, topography, traffic [cf. 23], and weather conditions. According to the ratings with the ACRES [25], the participants found that these environments ranged on average between 9.1–11.6 units in terms of inhibits or stimulates cycling (1 inhibits a lot; 8 being neutral; 15 stimulates a lot) (Table 3). These somewhat stimulating route environments should be seen in relation to the simple and standardized exercise laboratory environment, without the possibility of stimuli from outside. Thus, it can be concluded that the cycling commutes, compared to the laboratory exercise, had; (1) an overall higher level of external stimuli e.g. through traffic, (2) more natural components, including e.g. greenery and possibly water, and (3) more varying ambient conditions. So, how could these environmental differences have affected the present RPE results?

Already Pennebaker and Lightner [7] noted that symptoms of fatigue during exercise can be coupled to the degree of external stimuli. In their first experiment, they found that subjects who heard distracting street sounds via headphones during treadmill exercise reported less fatigue symptoms than subjects hearing an amplification of their own breathing. These findings were confirmed by their second experiment, in which subjects jogged a similar distance in two different environments, a cross-country path, and a monotonous lap course on a field. Although the cross-country jogging required more focus on the external environment than the lap course, higher speeds were demonstrated at comparable levels of fatigue symptoms. The findings of the present study can be related to Pennebaker and Lightner [7] in various ways. First of all, their effect of auditory stimuli can be related to the traffic situations that exposed the cycle commuters to varying ambient sounds. Traffic noise has been shown to adversely affect both cyclists and pedestrians [38–40], and Miedema [41] has shown that such forms of auditory impact can create high levels of annoyance, which might influence the perception of exertion.

Furthermore, the traffic situations in the present study demanded certain levels of visual and integrative attention. These distractions may have diminished the responsiveness to the internal sensations from the physical exercise, and thus led to a lower perceived exertion in the field compared to the laboratory environment [cf. 7].

Similarly, results by Ceci and Hassmén [8] and Mieras and colleagues [9] indicate that an increased level of external stimuli can reduce the perceived exertion, compared to less stimuli, at equal exercise intensities (cf. S4 Appendix). Both studies compared physiological and psychological responses to exercise in two different environments, a laboratory setting (with low external stimuli) and a field setting (with high external stimuli). The field trials in Ceci and Hassmén [8] took place in a green and blue recreational area, and the cycling in Mieras and colleagues [9] (cf. S2 Appendix) took place on a trail along a creek that in itself was a green and blue setting, while the overall framing setting for the trail was a mixture of primarily built up and green settings. Therefore, it is possible that the nature per se had an impact on the reduction of perceived exertion. The effect of soft fascination from nature and greenery on human wellbeing and restoration points in this direction [cf. 10, 11, 42]. In the present RPE assessments in field, the effect of greenery cannot, however, be isolated from the other commuting route environments, making it difficult to evaluate any potential effect of greenery on RPE. Therefore, further studies are needed to investigate this matter.

It is no doubt that the study by Ceci and Hassmén [8] involve the most green and beautiful external environment of the three studies using RPE measurements (cf. S1–S4 Appendices). At the same time, it is the study showing the greatest difference in physical performance. Whereas Mieras and colleagues [9] had 30% higher power outputs in field, the present study showed 19–30% higher relative levels of %HRR and %$\dot{V}O_2$max at given RPE levels, the average running speed Ceci and Hassmén [8] was 66% higher outdoors compared to indoors at given RPE levels (cf. S4 Appendix). This difference stimulates to a

hypothesis that greenery and aesthetics can possibly have a specific and greater role in this context, a matter that deserves future studies.

Another potential cause for the present higher RPE levels in the laboratory compared to the field setting could be due to that the participants experienced the cycling commuting somewhat unsafe, which demanded directed attention. The participants' ACRES ratings for unsafety-safety traffic of the overall route environments ranged on average between 5.6–10.9 (1 very unsafe; 8 neutral; 15 very safe) (Table 3). Consequently, a traffic-related stress could have caused an elevated HR relative to the workload and $\dot{V}O_2$ demand. This in turn could have led to the rated RPE levels in the field being compared with the laboratory based predicted RPE levels at a too high %HRR intensity. Contradictory to this, however, the relationships between %$\dot{V}O_2$max and RPE also showed significant RPE differences between the two conditions. Moreover, this theory is further refuted by the fact that relationships between HR and $\dot{V}O_2$ established during ergometer cycling in the laboratory have been shown to be valid for estimating intensity spectrums of $\dot{V}O_2$ based on HR measurements during cycle commuting in field for the same group of participants as in the current study [43].

Finally, the environmental differences in ambient conditions are also worth mentioning. In this respect, the average temperature during the cycle commuting was 10 and 12°C for males and females (Table 2), respectively, while in the laboratory it was kept around 20°C. Regarding this rather narrow temperature range, there is, to our knowledge, no study that clearly points in the direction of the present RPE results. On the contrary, Maw and colleagues [44] showed that when cool (8°C) and natural (24°C) temperature conditions were compared during constant ergometer cycling at the same absolute HR level corresponding to the currently used cycle commuting average (around 136 beats · min$^{-1}$; Table 7), the mean RPE difference between the two conditions was less than 0.5 RPE units.

## Applications and external validity

It can be beneficial from a health-promoting perspective that physical activities performed with the same exercise intensity are perceived as less strenuous outdoors than indoors. This perceived lower degree of exertion may lead to less physically active and sedentary individuals becoming more active and gain health effects.

At the same time, the findings in this study should be considered when prescribing exercise in a medical context. This is because classifications of exercise intensity by e.g. the American College of Sports Medicine (ACSM) [45] indicate that the currently used average commuting intensity (about 65% of $\dot{V}O_2$max; Table 7) corresponds to the lower part of the vigorous exercise intensity domain. According to the ACSM classification, this would correspond to an RPE level of approximately 14, which agrees very well with the laboratory based predicted RPE levels in this study (Table 8). However, the present rated RPE levels in the field (11.5–12.6; Table 8) fall between light to moderate exercise intensities according to this classification [45]. Supporting these findings on cycle commuting is a recent study of walking commuting [46] in which correspondingly lower RPE values were noted during field walking compared to what would be expected based on the relations between RPE and levels of %HRR and %$\dot{V}O_2$max stated by ACSM [cf. 45]. Given this, an RPE instructing exercise prescription that has been based on a relationship between perceived exertion and exercise intensity indoors risks leading to an elevated intensity level when applied outdoors. This risk should be especially considered when exercise is instructed for individuals with impaired health, e.g. due to heart disease [cf. 8].

With both lower and higher exercise intensities than applied in this study, the balance between internal and external cues may very well differ. It is, for example, reasonable to assume that increasing the relative exercise intensities might lead to that the internal cues

gradually dominate more and more over the external cues. Thereby the masking effect of external cues might decrease to a point where there are no effects left by them. This is supported by the findings of Ceci and Hassmén [8], in which the relative differences in running speeds decreased with higher RPE levels used to produce the running speeds (see S4 Appendix). Therefore, we do not believe that the present RPE differences observed between the different environmental conditions might be the same at both lower and higher exercise intensities. Indeed, we suggest that this issue is further studied.

## Strengths and limitations

It is a strength that the participants in this study were selected to be representative of a larger group of active commuters [12], in terms of sex, age, commuting mode, and distance. Thus, the present results should be considered valid in other habitual middle-aged commuters who cycle in inner urban and suburban areas. Whether the same applies to other groups of individuals such as young adults, the elderly, and athletes needs further investigation.

The present results are strengthened by the fact that the RPE assessments in the laboratory were demonstrated to be stable over time. In analyses of the 14 participants who performed an extra, third, cycle ergometer test in the laboratory, no significant differences were noted in any of the RPE comparisons with the second laboratory test (range of all mean differences: -0.7 -0.2 RPE units) (Table 5). This is in line with the high reliability previously observed for RPE assessments in both laboratory and field conditions [8].

It is a strength that both "central" (breathing) and "local" (leg muscles) RPE were assessed. The fact that similar results were noted in these two variables strengthen the findings and indicates that a general phenomenon may underlie the differences in RPE between ergometer cycling in a laboratory and commuter cycling in urban and suburban commuting environments.

Valid and well controlled measurement equipment was used in both the laboratory and field conditions [15–17]. This enabled that predicted RPE levels based on linear regression equations from ergometer cycling data in a laboratory could, for the first time, be compared with ratings of RPE from field cycling at equal exercise intensities. Simultaneously as this comparison is valuable, the fact that it makes use of two different forms of RPE assessments (ratings and predictions) is not optimal. Initially, a generalization is made when linear regressions are created between RPE and the average exercise intensities (%HRR and %$\dot{V}O_2$max) from the three cycle ergometer workloads. Subsequently, when these regression equations are used to predict the laboratory based RPE levels, several potential moderating factors may have been incorporated into the comparisons with the rated RPE values in the field.

The fact that neither cycling cadence nor power output was measured during the cycle commutes in the field is a limitation. These variables had made it easier to understand the physical work performed in the field and its relation to the ergometer cycling in the laboratory.

Additionally, the interpretation of results is limited by the fact that the two conditions used different test protocols and instructions. For instance, it would have been more optimal for this study to first perform the cycle commuting test in field and then a test with continuous cycling in the laboratory with the same exercise duration and average relative intensity as during the field cycling.

## Conclusion

The present study extends the previous research demonstrating that perceived exertion induced from aerobic exercise can be related to the degree of external stimuli from the

environment. In this case, it was illuminated by cycle commuters who assessed significantly higher RPE when performing ergometer cycling in the laboratory versus cycling commuting in suburban-urban environments at equal exercise intensities. Although several moderating factors (such as cycling cadence and exercise character) may have influenced the results, they are in line with previous studies pointing to external cues being modifiers of RPE. The findings prompt further studies of these matters.

## Supporting information

**S1 Appendix. Environmental description of the study by Ceci & Hassmén 1991.**
(PDF)

**S2 Appendix. Environmental description of the study by Mieras et al. 2014.**
(PDF)

**S3 Appendix. Environmental description of the present study by Olsson et al. 2024.**
(PDF)

**S4 Appendix. A comparison of four studies related to physical activity, perceived exertion and environment.**
(PDF)

**S5 Appendix. Comparisons of average exercise intensities between the entire cycle commutes versus the last 5-minute periods of the cycle commutes.**
(PDF)

**S1 Checklist. *PLOS ONE* clinical studies checklist.**
(DOCX)

## Acknowledgments

The authors are grateful to the volunteers for participating in the study, and for the technical assistance of Phoung Pihlträd, Jane Salier Eriksson, Cecilia Schantz-Eyre, Per Brink, Golam Sajid, Eva Minten and Erik Stigell. Drs James Pennebaker, Ruggero Ceci, Peter Hassmén and Dustin Slivka are thanked for sharing information about both the laboratory and environmental settings used in their studies. Landscape architect Dennis Bryers at Omaha Parks, Recreation & Public Property Department, Omaha, Nebraska, USA, is thanked for his generous assistance in sharing information and photos of Keystone Trail. Michael Schonlau, GIS Administrator for the Douglas County GIS Department, Nebraska, USA, and Hans-Olov Andersson, at The Land Survey, Gävle, Sweden, are thanked for assistance with aerial photos. Dr Magnus Strömgren is thanked for creating and letting us use a map. Finally, we express our gratitude to the reviewers for valuable comments.

## Author Contributions

**Conceptualization:** Karin Sofia Elisabeth Olsson, Peter Schantz.

**Data curation:** Karin Sofia Elisabeth Olsson, Lina Wahlgren, Hans Rosdahl, Peter Schantz.

**Formal analysis:** Karin Sofia Elisabeth Olsson, Ruggero Ceci, Peter Schantz.

**Funding acquisition:** Peter Schantz.

**Investigation:** Ruggero Ceci, Lina Wahlgren, Peter Schantz.

**Methodology:** Karin Sofia Elisabeth Olsson, Hans Rosdahl, Peter Schantz.

**Project administration:** Lina Wahlgren, Peter Schantz.

**Supervision:** Peter Schantz.

**Validation:** Hans Rosdahl, Peter Schantz.

**Visualization:** Karin Sofia Elisabeth Olsson.

**Writing – original draft:** Karin Sofia Elisabeth Olsson, Ruggero Ceci, Peter Schantz.

**Writing – review & editing:** Karin Sofia Elisabeth Olsson, Ruggero Ceci, Lina Wahlgren, Hans Rosdahl, Peter Schantz.

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
