## [Decision Letter · Decision Letter 0]

5 Jun 2023

PONE-D-23-12379Perceived exertion is lower at equal exercise intensities when cycling in field versus indoorPLOS ONE

Dear Dr. Schantz,

Thank you for submitting your manuscript to PLOS ONE. After careful consideration, we feel that it has merit but does not fully meet PLOS ONE’s publication criteria as it currently stands. Therefore, we invite you to submit a revised version of the manuscript that addresses the points raised during the review process. As it can be gathered from the comments below, the reviewers have somewhat different levels of enthusiasm about the overall value of the data (please check attachment with comments from Reviewer #1). Nevertheless, based on the reports, I would like to invite the authors to respond to the comments from each reviewers.

We look forward to receiving your revised manuscript.

Kind regards,

Juan M. Murias

Academic Editor

PLOS ONE

3. e note that you have included the phrase “data not shown” in your manuscript. Unfortunately, this does not meet our data sharing requirements. PLOS does not permit references to inaccessible data. We require that authors provide all relevant data within the paper, Supporting Information files, or in an acceptable, public repository. Please add a citation to support this phrase or upload the data that corresponds with these findings to a stable repository (such as Figshare or Dryad) and provide and URLs, DOIs, or accession numbers that may be used to access these data. Or, if the data are not a core part of the research being presented in your study, we ask that you remove the phrase that refers to these data.

Reviewers' comments:

Reviewer's Responses to Questions

**Comments to the Author**

1. Is the manuscript technically sound, and do the data support the conclusions?

Reviewer #1: No

Reviewer #2: Yes

2. Has the statistical analysis been performed appropriately and rigorously? 

Reviewer #1: No

Reviewer #2: Yes

3. Have the authors made all data underlying the findings in their manuscript fully available?

Reviewer #1: Yes

Reviewer #2: No

4. Is the manuscript presented in an intelligible fashion and written in standard English?

Reviewer #1: No

Reviewer #2: Yes

5. Review Comments to the Author

Reviewer #1: Dear authors,

I must commend the amount of work done for the current manuscript. I much appreciated reading the material. However, I think there are some topics, specifically in the methods, that require some thoughts and updates to make it more clear. Hope my commentaries are of some value.

Best regards

Reviewer #2: The authors aimed to assess whether cycling outside at a given intensity led to lower perceived exertion than cycling in a laboratory setting. They present interesting findings that suggest that cycling outside during a commute leads to lower RPE at a given intensity. I commend the authors on their analysis, for example the assessment of the stability of RPE in the laboratory. I believe their data is interesting. I include some revisions below, which are mostly about things I would like to see discussed in a bit more depth in the manuscript.

Introduction:

Line 52: instead of founder maybe creator or something similar would be a better word?

Line 81: This seems to suggest that for a given RPE participants were able to run 80% faster outdoors, i.e. 10 vs 18 km/h.Please clarify.

Line 109: If external stimuli mask internal cues from physical exercise, wouldn’t it be a better option to assess this phenomenon to use a high intensity exercise? Or at least an exercise intensity that requires some level of attention to internal cues, instead of the relatively low intensity required to commute, especially if it is a commute that is repeated dozens to hundreds of times, in which it is very easy to “turn off” the internal perception because it is simply the same as every other time.

Line 113-126. Is this not the study design? As such, shouldn't it be the first one in methods?

Methods:

Why did the authors not use the same metabolic cart for both tests? The portable one could have been used during the laboratory test as well. Do the authors have any comparison between their metabolic carts?

How stable were VO2 and heart rate, considering that there could have been several stops and changes of direction and speed during the commute? This makes me wonder whether VO2 and HR are good measures of intensity for commuting, given the fact that they may not reach steady state. However, I do not think this invalidates the results. If anything, a lack of steady state would mean that the effort was actually higher than the VO2 or HR reflect, and RPE should be higher, and not lower, so the result is still valid and very interesting.

This would be unless the distribution of effort was not uniform. For example, the men commuted at a VO2 which was almost equivalent to 193 W in the lab, and 65% of VO2max. For the men, that is equivalent to an average 2.27 W/kg, which considering that the average BMI for the men was 24.7 and their age, I am assuming that they had a relatively high fat mass. For example, if we calculate the BMI for the same height but 10 kg less (75 kg) it is 21.9,, which would still not be extremely lean. A lean 75 kg man, riding at 193 W, would be riding at 2.57 W/kg. In my experience, those are common training intensities for endurance exercise for well trained amateur cyclists, and also close to the limit of FTP or other measures of the second lactate turnpoint in trained cyclists of lower level. Therefore, it seems strange to me that the intensity they were sustaining was hard enough to bring them to those power outputs. This makes me wonder whether, instead of keeping that steady state intensity, the commuter may have been cycling in bursts of high intensity interspersed with lower intensity cycling, which could maybe have kept VO2 high on average, but RPE lower, due to the recovery time at lower intensity or during stops. Furthermore, those power outputs of approximately 190W would have been sustained, to give that average VO2max, as constant intensity exercise. Taking into account the intermittent nature of commuting, I suspect that they would have had to sustained pretty high power values to reach that VO2max, which I do not think is usually done by commuters, especially on their way to work, since they don’t want to get there sweaty and tired.

Line 289: RPE was asked as the RPE for the whole commute, vs RPE in the moment during pedaling. Have the authors considered the possible effect of the Peak-End rule? (https://psycnet.apa.org/doiLanding?doi=10.1037%2F0022-3514.65.1.45) Being asked after finishing to rate the RPE for the whole commute could be confounded by the cyclists using the peak-end rule and averaging only the highest RPE they felt and theRPE at the end.

Discussion:

In line 505 it is suggested that those who cycle longer durations have higher VO2max. Does that hold true in your data set?

Line 544: Again, during a commute, the average VO2max or HR may not be good measures of intensity, considering the amount of starting and stopping likely to be present. For example, if one 5 minute period was continuous cruising and the other one had a total of 1 min of rest divided in 4 stops, or a series of uphill climbing and downhill freewheeling, the second one would be much harder. This would be reflected as higher lactate, for example, even if VO2max was similar.

L 614-622. Regarding environmental factors, what was the ventilation during exercise in the laboratory? Cycling at 20ºC in still air can produce an increasing level of discomfort as body temperature rises, which would increase the perceived difference of temperature between the lab and the field, and I suspect this could have a significant impact on RPE.

Generalising the results across intensities? Do the authors believe that these results can be generalised across intensities? In my experience, at higher intensity and for a given RPE, having to integrate more information leads to lower exercise intensities and lower speeds. For example, having to perform cycling intervals in somewhat heavy traffic makes cyclists ride slower, at lower average power and with more variations in power, even when they consider that the effort was equally hard.

6. PLOS authors have the option to publish the peer review history of their article (what does this mean?). If published, this will include your full peer review and any attached files.

Reviewer #1: **Yes: **Rafael de Almeida Azevedo

Reviewer #2: No

---

## [Author Response · Author response to Decision Letter 0]

17 Jul 2023

Response to the editor and the reviewers

Many thanks for all the work undertaken by you. The comments have been clearly helpful in advancing the quality of the ms.

We have undertaken one change that was not requested by the reviewers; a modification in Figure 3, excluding the regression lines that were based on all individuals’ values. The reason for this was to diminish the risk for misinterpretations of our main analysis strategy, as is visualized in Figure 1. 

Each of our responses are given after three asterisks *** 

*** This has now been checked.

2. We note that you have indicated that data from this study are available upon request. PLOS only allows data to be available upon request if there are legal or ethical restrictions on sharing data publicly. 

*** The underlying research materials related to this paper are not freely and directly available because the original approval by the ethics board (The Ethics Committee North of the Karolinska Institute at the Karolinska Hospital, Stockholm, Sweden (Dnr 03- 637)) and the informed consent from the participants do not include such direct free access. The data will be available to all interested researchers upon request. To gain access to the data, please contact the Registrator at The Swedish School of Sport and Health Sciences, GIH, Box 5626, SE-114 86 Stockholm, Sweden, tel: +46 (0)812053700, email: registrator@gih.se.

3. e note that you have included the phrase “data not shown” in your manuscript. Unfortunately, this does not meet our data sharing requirements. PLOS does not permit references to inaccessible data. We require that authors provide all relevant data within the paper, Supporting Information files, or in an acceptable, public repository. Please add a citation to support this phrase or upload the data that corresponds with these findings to a stable repository (such as Figshare or Dryad) and provide and URLs, DOIs, or accession numbers that may be used to access these data. Or, if the data are not a core part of the research being presented in your study, we ask that you remove the phrase that refers to these data.

*** This is now changed. Some data is reported in a Supporting Information file, named Appendix S1.

Responses to the authors

Reviewer #1:

Dear authors,

I must commend the amount of work done for the current manuscript. I much appreciated reading the material. However, I think there are some topics, specifically in the methods, that require some thoughts and updates to make it more clear. Hope my commentaries are of some value.

*** Many thanks for your comments.

General commentaries 

The present study aimed to investigate the relationship between indoor vs outdoor cycling RPE in habitual commuters. In order to establish this relationship, the authors collected metabolic data (HR and VO2) after a step incremental test, performed indoor, and assessed the averaged RPE response after a outdoor cycling. The results showed that the predicted RPE, based on indoor cycling, overestimates the RPE shown after an outdoor cycling bout, when normalizing by the metabolic rate (HR and VO2). Even though the topic of the present manuscript is interesting to guide public governance in terms of active commuting policies, the present methods and results have serious limitations to address the proposed goal. Please, see the commentaries below for specific issues and details. 

Specific commentaries 

Abstract 

L.29: Since running was the example briefly discussed in the abstract, it would be better to introduce the cycling topic earlier. Some naïve reader could ask running itself was not studied, but cycling was the chosen modality, see l 20-30. 

*** Given that the study on running, and its environmental contexts, created the motives for this study, we think it is reasonable that it is presented as it stands in the background. Instead, we have modified the remaining text so that it connects better to the previous one. It now stands: It would therefore be valuable to explore whether the same applies while cycling in a laboratory versus in cycle commuting environments with high levels of stimuli from both traffic and suburban-urban elements. That is the aim of this study.

L. 33-34: from the abstract it is not clear the reason of not using dyspnea scale for breathing. Instead, the authors utilized RPE (6-20) in order to assess breathing and legs. 

*** We did not anticipate that the submaximal work rates in the laboratory and in field cycling would involve dyspnea, and therefore such a scale was not considered.

L. 38: from the abstract, it is not clear how exercise intensity was assessed outside of the lab.

*** Many thanks for this point. It has now been added that we used portable monitoring systems (p. p 2, line 32). 

Introduction 

L. 51-56: Even though I agree about the information provided in this paragraph, it must be acknowledged that the actual methods to establish the RPE – HR relationship was the % of HR reserve. So, it would be reasonable to at least mention that %HRreserve could also be utilized. 

*** Thanks for this point. We have now added that “the relations to both absolute and relative HR levels remain”.

L. 60: Note that the authors mentioned that RPE is a result of several ongoing signals, including breathing and muscle work. Thus, it would be interesting to discuss the reason for adopting RPE-breathing and RPE-leg utilized in the methods. 

*** Thanks for this point. We think a comment on this matter might be more suitable when we in “Methods" state our usage of this RPE division under the heading “Study design and standardization”. We have now added this text, l. 170-72: The reason for this division is that RPE for a given workload is higher when it is executed by a small vs a large muscle group, whereas the central RPE, referred to as breathing, can be similar. 

L. 62-66: Even though it is not the aim of the present study, someone could argue that RPE is also a result of feedforward based on expectations. For example, athletes that perform a self-paced exercise in a closed-loop task always pace themselves, partially based on the expectation to reach the finish line. It could be acknowledged here that not only afferent signals, but also some supraspinal mechanisms. Some information on the topic is given in the paragraph just below, maybe they could be merged. Please, see this meta-analysis for more details on this topic: Pharmacological Blockade of Muscle Afferents and Perception of Effort: A Systematic Review with Meta-analysis; DOI: 10.1007/s40279-022-01762-4 

*** Good point! Thanks. Indeed these complementary perspectives are of clear value to mention. We have therefore added the following text, l. 64-68: On the other hand, there are indications that at least afferent feedback from the working muscles to the brain may not play a role in generating the perception of effort (Bergevin et al. 2023). In line with this, a contrasting theory to the peripheral origin of effort has been developed; it suggests that the perception of exertion might originate from a copy of the central motor command, a so-called “corollary discharge” (cf. Bergevin et al. 2023). 

L. 68-73: Some studies have utilized opponents to assess the effect of external clues on self-paced exercise intensity. Please, see the following reference: Deception Improves Time Trial Performance in Well-trained Cyclists without Augmented Fatigue; DOI: 10.1249/MSS.0000000000001483

*** Thanks for this article, widening the potential importance of external cues in altering maximal performance levels. Given that we focus on a submaximal efforts, we will, however, not refer to it. 

L. 94: I assume that “three later studies” is referring to refs# 1, 7 and 9. However, only refs # 7 and 9 were explained. Would it be better to just say “given these abovementioned studies”? 

*** We have modified and clarified this now.

L. 94-102: I see the relevance of the study, however previous data on cycling performance has already investigated the effect of external clues on self-selected exercise intensity. Please, see the following refences: Listening to Music in the First, but not the Last 1.5 km of a 5-km Running Trial Alters Pacing Strategy and Improves Performance, DOI http://dx.doi.org/10.1055/s-0032-1311581; AND Time to move beyond a brainless exercise physiology: the evidence for complex regulation of human exercise performance, doi:10.1139/H10-082 

*** Thanks again for interesting studies! They are definitely studies that are related to our study area, but the present aim was not to investigate the effect of external, auditory, clues on maximal exercise performance. 

L. 100-103: Since the researchers aimed to investigate the effect of environmental clues on RPE and self-regulation of exercise intensity, I would recommend revising the references utilized in the introduction. For example, refs 11 and 10 are mostly related to surgery recovery, which is a completely difference scenario when it comes to active commuting on a bicycle. 

*** Just to clarify; we aim at investigating “the effect of environmental cues on RPE”, but, we have no focus on “self-regulation of exercise intensity”. We agree that it is reasonable to omit ref. 11 in relation to stress reduction but will let it remain as a “confer”-source to other studies describing preferences and positive psychological effects.

L. 121: It is not clear the reason for utilizing % of VO2 and %HR reserve, since borg scale is set to be comparable to absolute HR responses. 

*** It is true that the Borg scale was, initially developed in relation to HR. Later, however, even relative HR measures were used (Borg 1998, see the ref. list.). This is important, since HR is related to both % of VO2 and % of HR reserve, those variables can also be used. Furthermore, these variables consider e.g. interindividual differences in maximal HR and training status. Therefore, in our mind, they fit better to be matched with the RPE scale. We have added comments on this in the ms, see l. 350-351.

L.113-126: This paragraph is methodologically right, but it does not provide the main goal and hypothesis of the current manuscript. I would strongly suggest the authors to rewrite in order to make their claim stronger and show the relevance of the work done. 

*** Point is taken, please note the modified version of the last paragraph in Introduction. See the text on; l. 115-123.

For example, the ACRES point was not explored in the introduction. To me, this is a key aspect of the present work. 

*** Using ACRES is a novel dimension, but still we have decided to not introduce it in the revised version of the Introduction.

Also, have the participants rated the positive and negative appraisals for the lab environment as well? 

*** No. However, we secured that the laboratory environment was low on visual stimulation given that the participants looked at a uniform shield, but the perceptions evoked by it were not scored. This is now described in the Methods (l. 278-279). 

This is relevant since some people prefer cycling indoors because of better environmental control. Additionally, why only including habitual cycle commuters? This seems to be a bias in the present sample, since those participants are used to outdoor cycling but maybe less inclined for indoor cycling. 

*** We see clear advantages with our strategy of using experienced cyclists, but, as often is the case, the external validity of the findings can be discussed, and prompts further studies of these matters. We think that your reflections are covered in the first paragraph under the heading “Strengths and limitations” in the Discussion.

Methods

L. 143: In the introduction it was stated that the participants were “habitual cyclists”, however, the inclusion criteria was “at least once a year”. How habitual cyclists was defined for the study?

*** We have now added data into Table 1 demonstrating that the participants are habitual cyclists. This is illuminated by their high levels of commuting frequency and total cycling distance per year. Furthermore, we have now also clarified that the criteria used to select the present sample were based on the overall project’s median values of the male and female single mode cyclists. (Page 5, line 148) 

Table 1: It would be of great value for the reader to see the frequency and distance cycled throughout the participant’s commuting route. This would provide more details about the sample here utilized. 

*** Thanks for the suggestion. We agree. Cycling frequency (number of trips per year) and distance cycled per year are now described in Table 1. Cycled distance per commuting trip is already described in the results section in Table 6.

L. 167: Two or three sessions depending on which variable?

*** The explanation for why some participants did two test sessions and others three is given on page 6, lines 177-180.

L. 171-173: The difference between RPEs here is not clear. Based on Borg’s work, RPE would be a general scale not specific to a given limb or sensation. Additionally, there are specific scales about muscle pain, discomfort and/or dyspnea that would be more useful. Has the authors asked only RPE, without specifying the sensation or limb, to the participants?

*** We agree that Borg's RPE 6-20 scale was originally intended to assess the general perceived exertion. However, it is common practice at some laboratories, at least in Sweden, that the scale is used to assess the perceived exertion for breathing and musculature separately. This has long been practiced, see background in, e.g.; 

- Ekblom B, Goldbarg AN. The influence of physical training and other factors on the subjective rating of perceived exertion. Acta Physiol Scand. 1971;83(3):399-406.

- Borg G. Borg's perceived exertion and pain scales. Champaign, IL: Human Kinetics Publishers; 1998.

The participants were instructed to rate their perceived exertion for breathing and the leg musculature separately. We consider it advantageous that the same scale numbering has been used for assessment of two different sources of perceived exertion. In addition, unlike pain scales and the dyspnea scale, the RPE 6-20 scale is well suited for measuring aerobic exercise over a wide intensity range.

Figure 2: The figure was not displayed, so I cannot comment on the quality of the data.

*** This is the figure that was submitted.

Figure 3: Could the authors improve figure’s 3 resolution? It is quite hard to understand the letter and values shown there. 

*** The resolution of the submitted figure is 600 dpi. Sorry that it did not appear in the review version. The modified figure is enclosed below.

L. 200-216: It is unclear the reason for not using the same metabolic cart for indoor and outdoor cycling. What is the bias of using those two different systems? Would that interfere in the results here presented? 

*** A very reasonable point. We, however, checked that these two systems do not differ through using the same metabolic calibrator. This is stated in the second paragraph under the heading “Metabolic systems”, and further described in the cited work; Schantz et al. 2018. A sentence stating the issue of interchangeability of the systems is now added to the text, l. 214-215: “This included checks with a metabolic calibrator to ensure that the two systems were interchangeable.” 

L. 248-249: How steady state HR was defined here? Based on a given number over X minutes?

*** This has now been made clear in the script, see page 7, line 252-253: Steady state HR was defined as when HR values were within 2-3 beats · min-1, during two consecutive min

---

## [Decision Letter · Decision Letter 1]

17 Aug 2023

PONE-D-23-12379R1Perceived exertion is lower at equal mean exercise intensities when cycling in field versus indoorsPLOS ONE

Dear Dr. Schantz,

Thank you for submitting your manuscript to PLOS ONE. After careful consideration, we feel that it has merit but does not fully meet PLOS ONE’s publication criteria as it currently stands. Therefore, we invite you to submit a revised version of the manuscript that addresses the points raised during the review process.

As you will see in the document that one of the reviewers attached, some minor but important revisions are requested. I trust that no further interactions will be needed once these points are addressed.

We look forward to receiving your revised manuscript.

Kind regards,

Juan M. Murias

Academic Editor

PLOS ONE

Journal Requirements:

Reviewers' comments:

Reviewer's Responses to Questions

**Comments to the Author**

1. If the authors have adequately addressed your comments raised in a previous round of review and you feel that this manuscript is now acceptable for publication, you may indicate that here to bypass the “Comments to the Author” section, enter your conflict of interest statement in the “Confidential to Editor” section, and submit your "Accept" recommendation.

Reviewer #1: All comments have been addressed

2. Is the manuscript technically sound, and do the data support the conclusions?

Reviewer #1: Partly

3. Has the statistical analysis been performed appropriately and rigorously? 

Reviewer #1: Yes

4. Have the authors made all data underlying the findings in their manuscript fully available?

Reviewer #1: Yes

5. Is the manuscript presented in an intelligible fashion and written in standard English?

Reviewer #1: Yes

6. Review Comments to the Author

Reviewer #1: Dear authors,

Much appreciate all the effort and attention in responses. However, I do have very minor points, which do not require major changes but are worthy to be discussed. Please, see the document attached.

7. PLOS authors have the option to publish the peer review history of their article (what does this mean?). If published, this will include your full peer review and any attached files.

Reviewer #1: No

---

## [Author Response · Author response to Decision Letter 1]

16 Feb 2024

Authors’ response to the editor and the two reviewers in conjunction with resubmission 2 

In working with modifications given comments by reviewer no 1, matters developed. 

First of all; we thought that we should add one analytical step to make our study more comparable with three other studies in the field. It now is presented in Table 9, and shows that the calculated increase in physical exercise intensity at a given RPE appears to be between 19-30% when field exercise is compared with indoor laboratory conditions. 

Furthermore, we felt that now is the time to further these matters with regard to a specific weakness in previous related studies, namely the descriptions of the laboratory and external environments used. 

We have therefore interviewed all corresponding authors of the previous related studies to clarify the external environmental characteristics, and to describe these issues in Supplementary Appendix files. We also create an overview of all 4 studies in an additional file. 

These matters do not change any principal issues in comparison with our initial submission, but they strengthen it considerably. 

All changes undertaken are shown as either “track changes”. 

For specific comments to reviewer no 1´s response after resubmission 1, see below.

Authors’ response to reviewer no 1 response after resubmission 1

Many thanks for your further fruitful comments and recommendations that we highly appreciate. 

Our comments are, given that, stated at: ***

Dear authors,

Much appreciated all the effort and attention in your response. I feel that the manuscript improved, and hope the authors have the same opinion. 

Nonetheless, I do have some minor commentaries (in red), which do not require changes. These are only recommendations. 

Reviewer: L. 33-34: from the abstract it is not clear the reason of not using dyspnea scale for breathing. Instead, the authors utilized RPE (6-20) in order to assess breathing and legs. 

Authors: *** We did not anticipate that the submaximal work rates in the laboratory and in field cycling would involve dyspnea, and therefore such a scale was not considered. 

Reviewer: Ok, I guess these differences were explained in the new version, but it would be great including even more details about RPE underpinning mechanisms during exercise, with special attention to breathing and specific muscle group (legs in this case). 

*** First, just a further comment stressing why we have not made use of a dyspnea scale. Based on previous research of cycle commuting (Oja et al. 1991, se ref. in Schantz et al. 2022. Frontiers in Public Health 10(911863):1-19), it could not be expected that any dyspnea would occur in our study. The results were also in line with that. The mean exercise rates in the current field cycling, undertaken of healthy participants, were about 65% of maximal oxygen uptake, and are normally not evoking any hyperventilation. The ventilatory equivalents were in line with that. Therefore, we think that introducing aspects related to the dyspnea scale would not contribute to furthering the understanding of the main aim with this study.

Regarding “but it would be great including even more details about RPE underpinning mechanisms during exercise, with special attention to breathing and specific muscle group (legs in this case).”, we cannot provide any more such information than that already stated and discussed in the ms. Indeed, given the references in this respect, mentioned in the Introduction, we noted that this field seems to clearly warrant more research in future.

Reviewer: L. 60: Note that the authors mentioned that RPE is a result of several ongoing signals, including breathing and muscle work. Thus, it would be interesting to discuss the reason for adopting RPE-breathing and RPE-leg utilized in the methods. 

Authors: *** Thanks for this point. We think a comment on this matter might be more suitable when we in “Methods" state our usage of this RPE division under the heading “Study design and standardization”. We have now added this text, l. 170-72: The reason for this division is that RPE for a given workload is higher when it is executed by a small vs a large muscle group, whereas the central RPE, referred to as breathing, can be similar. 

Reviewer: Still, some information should be added in the limitations section of the study. 

*** Thanks. In our mind, given the comments above, we regard it to be a “strength” that we have both these kinds of RPE-measurements, and that they point in the same direction. We have added the following comment on this matter under the heading “Strengths and limitations” in the Discussion: “It is a strength that both “central” (breathing) and “local” (leg muscles) RPE were assessed. The fact that similar results were noted with these two measures strengthen the findings and point to that a general phenomenon might be underlying the differences in RPE between stationary cycling in laboratory and commuter cycling in urban and suburban commuting route environments.” 

Reviewer: L.113-126: This paragraph is methodologically right, but it does not provide the main goal and hypothesis of the current manuscript. I would strongly suggest the authors to rewrite in order to make their claim stronger and show the relevance of the work done. 

Authors: *** Point is taken, please note the modified version of the last paragraph in Introduction. See the text on; l. 115-123. 

Reviewer: Thanks for the update. As a suggestion, the figure 1 should be only mentioned once in the methods section not in the introduction. 

*** We think that this illustration is clearly important for easily understanding the analytical approach used. Therefore, it is of clear value that it is found early in the ms. Given that we think that it deserves a placing in the Introduction.

Reviewer: Also, have the participants rated the positive and negative appraisals for the lab environment as well? 

Authors: *** No. However, we secured that the laboratory environment was low on visual stimulation given that the participants looked at a uniform shield, but the perceptions evoked by it were not scored. This is now described in the Methods (l. 278-279). 

Reviewer: It would be great if the participants were previously familiarized with lab environment, thus minimizing any external clue. In case you did a familiarization, this would be the place to cite it. 

*** Yes, they were. And we have now added that aspect after the sentence starting at l. 174: It reads: “The first test of ergometer cycle exercise in the laboratory was carried out with a purpose of familiarization”. Now it continues with: “, with regard to both all aspects of the testing, and to the laboratory environment.”

Reviewer: L. 171-173: The difference between RPEs here is not clear. Based on Borg’s work, RPE would be a general scale not specific to a given limb or sensation. Additionally, there are specific scales about muscle pain, discomfort and/or dyspnea that would be more useful. Has the authors asked only RPE, 

without specifying the sensation or limb, to the participants? *** We agree that Borg's RPE 6-20 scale was originally intended to assess the general perceived exertion. However, it is common practice at some laboratories, at least in Sweden, that the scale is used to assess the perceived exertion for breathing and musculature separately. This has long been practiced, see background in… 

Reviewer: I would argue that the relevance of internal sensations induced by exercise, such as exertion, pain and dyspnea, may vary according to the intensity domain. Thus, RPE should not be used as an universal measurement of internal load during exercise. 

See the manuscript below: https://doi.org/10.1152/japplphysiol.00764.2021

*** We fully agree with you on this matter regarding a spectrum of physical activities and exercises, but given that the overall exertion in cycle commuting is rather low, we think that for our purposes RPE is a sufficient measure.

---

## [Decision Letter · Decision Letter 2]

6 Mar 2024

Perceived exertion can be lower when exercising in field versus indoors

PONE-D-23-12379R2

Dear Dr. Schantz,

We’re pleased to inform you that your manuscript has been judged scientifically suitable for publication and will be formally accepted for publication once it meets all outstanding technical requirements.

Kind regards,

Juan M. Murias

Academic Editor

PLOS ONE

Additional Editor Comments (optional):

Reviewers' comments:

Reviewer's Responses to Questions

**Comments to the Author**

1. If the authors have adequately addressed your comments raised in a previous round of review and you feel that this manuscript is now acceptable for publication, you may indicate that here to bypass the “Comments to the Author” section, enter your conflict of interest statement in the “Confidential to Editor” section, and submit your "Accept" recommendation.

Reviewer #1: All comments have been addressed

2. Is the manuscript technically sound, and do the data support the conclusions?

Reviewer #1: Yes

3. Has the statistical analysis been performed appropriately and rigorously? 

Reviewer #1: Yes

4. Have the authors made all data underlying the findings in their manuscript fully available?

Reviewer #1: Yes

5. Is the manuscript presented in an intelligible fashion and written in standard English?

Reviewer #1: Yes

6. Review Comments to the Author

Reviewer #1: Dear authors,

Congratulations for the work and manuscript. Hope this revision process enabled new ideas for future studies.

Best regards

7. PLOS authors have the option to publish the peer review history of their article (what does this mean?). If published, this will include your full peer review and any attached files.

Reviewer #1: No

---

## [Editor Report · Acceptance letter]

29 Apr 2024

PONE-D-23-12379R2 

PLOS ONE

Dear Dr. Schantz, 

I'm pleased to inform you that your manuscript has been deemed suitable for publication in PLOS ONE. Congratulations! Your manuscript is now being handed over to our production team.

Kind regards, 

on behalf of

Dr. Juan M. Murias 

Academic Editor

PLOS ONE